# Gait Speed and Sleep Duration Is Associated with Increased Risk of MCI in Older Community-Dwelling Adults

**DOI:** 10.3390/ijerph19137625

**Published:** 2022-06-22

**Authors:** Eunju Yoon, Seongryu Bae, Hyuntae Park

**Affiliations:** 1Department of Food Science and Nutrition, Dong-A University, Busan 49315, Korea; ejyoon@dau.ac.kr; 2Department of Healthcare and Science, Dong-A University, Busan 49315, Korea; srbae@dau.ac.kr

**Keywords:** gait speed, sleep, mild cognitive impairment, elderly

## Abstract

This study aimed to examine the linear and nonlinear associations between sleep duration and gait speed and the risk of developing mild cognitive impairment (MCI) in community-dwelling older adults. Participants were 233 older adults who met the study inclusion criteria. The MCI diagnosis was based on medical evaluations through a clinical interview conducted by a dementia specialist. Self-reported sleep duration was evaluated using the Pittsburgh Sleep Quality Index. The usual gait speed was calculated from the time taken to walk along a 4 m walkway. Multivariate logistic regression analysis was used to calculate the odds ratio (OR) and the 95% confidence interval (95% CI) of developing MCI in relation to sleep duration and gait speed. Generalized additive models were used to examine the dose–response relationships between sleep duration, gait speed, and the risk of developing MCI. Slower gait speed (OR: 1.84, 95%; CI: 1.00–3.13) and poor sleep duration (OR: 1.76, 95%; CI: 1.00–3.35) were associated with the risk of developing MCI, compared with their optimal status. In addition, the combination of poor sleep and slower gait was associated with a higher risk of developing MCI than optimal sleep duration and gait speed (OR: 3.13, 95%; CI: 1.93–5.14). Furthermore, gait speed and sleep duration were non-linearly associated with the risk of developing MCI. These results highlight the complex interplay and synergism between sleep duration and gait abilities on the risk of developing MCI in older adults. In addition, our results suggest that slower gait speed (<1.0 m/s) and short (<330 min) and long (>480 min) sleep duration may be linked to MCI risks through underlying pathways.

## 1. Introduction

Mild cognitive impairment (MCI) represents an early clinical stage of cognitive impairment that is distinct from normal aging, with the potential for further progression of Alzheimer’s disease or other dementias [1]. Predicting the earliest stages of cognitive impairment has important implications towards initiating treatment and monitoring disease progression.

Multi-modal modifiable factors, such as hypertension, sleep disturbance, physical dysfunction, physical inactivity, malnutrition, social contact, and physical and mental disorders, have been reported to be associated with dementia [2,3,4].

Slowing of motor function is commonly observed in elderly patients and may be more pronounced in older persons with cognitive impairment than in those who are cognitively intact [5,6,7]. Several studies have shown that slowing gait speed and/or motor slowing precedes and may predict the onset of cognitive impairment [8,9,10].

Sleep disturbances are common in the elderly, with as many as 50% of those aged over 65 years reporting a chronic sleep complaint. Many prospective cohort studies have examined the association between disturbed sleep and cognitive function in older adults, and reported that insomnia-type symptoms in community-dwelling older adults are associated with cognitive decline [4,11,12,13].

Moreover, longitudinal studies examining the relationship between slowing gait [6,9] on the one hand and the association between duration of sleep [4,12,13] at baseline and cognitive performance at follow-up on the other produced mixed results. Some reported that short or long sleep duration, or both, are associated with cognitive impairment, whereas other studies failed to document any such associations [4,14,15,16]. However, the time at which this slowing gait and changing sleep duration begins in relation to the onset of cognitive impairment is not clear. Moreover, the synergistic effects of gait speed and sleep on cognitive function in older adults remain unclear. Thus, this study aimed to examine the linear and nonlinear associations between gait speed and sleep duration and the risk of developing MCI in community-dwelling older adults.

## 2. Materials and Methods

### 2.1. Participants

In this cross-sectional study, our participants, i.e., community-dwelling older adults aged 65 or above, were recruited from four community health centers and institutions in Busan city, Korea.

We included a total 445 individuals who agreed to participate and who completed our screening survey. Of these 445 participants, 233 were selected for the study and 212 were excluded because of other disease diagnoses or a failure to meet the established criteria for inclusion. Reasons for exclusion included: adults under the age of 65 years; heart surgery; cardiovascular disease; heart failure; congenital heart disease; pacemaker attachment; under medication for cardiovascular or cerebrovascular diseases; currently under treatment for musculoskeletal disorders such as osteoarthritis or rheumatoid arthritis; experience of neurological abnormalities such as chest pain or numbness; vomiting during physical activity; diagnosed by a specialist with a specific disease and recommended to perform physical activity only under the guidance of a specialist, or prohibited from performing physical activity; reporting pain in the chest or arrhythmia when resting; missing data; and failure to agree to participate in the study in writing.

Among the final participants, 141 were women and 92 were men. Based on the methodology in a previously published paper [17], diagnostic assignment was made by the study’s neurologist based on data from the clinical examination, rating scales, and neuropsychological testing. With regard to the latter, subjects were classified as meeting criteria for amnestic MCI if they performed at least 1.5 standard deviations (SD) below the age/intelligence quotient-adjusted mean on at least 1 measure of memory functioning, with preserved general cognitive functioning.

All participants provided written informed consent and the study procedure was approved by the Institutional Review Board (IBR-BR-009-019).

### 2.2. Sleep Duration

Questions regarding sleep patterns (i.e., sleep duration and sleep timing) were adapted from the Pittsburgh Sleep Quality Index [18]. Sleep duration was investigated with the following question: “During the past month, how many hours of actual sleep did you get per night?”. The appropriate range of sleep duration is 5–9 h per day in older adults to maintain or improve their health [19]. The cut-off value of 330–480 min for optimal sleep duration was defined as the point at which MCI risk begins to increase from zero, based on our results of the GAM analysis.

Thus, the participants were divided into three groups according to their sleep duration: short, <330 min; optimal, 330–480 min; and long, >480 min. This result was consistent with the previous study which reported that individuals having ≤5 h or >8 or 9 h of sleep showed an increased risk of developing dementia or MCI [20,21].

### 2.3. Physical Activity, Gait Speed, and Physical Function

Daily physical activity was objectively measured through moderate-to-vigorous physical activity (MVPA) of intensity greater than 3 metabolic equivalents (METs). Measurements were recorded with a tri-axial accelerometer (Fitmit INC, Suwon, Korea). The accelerometer was worn on the non-dominant hand by the participants.

Details of the gait speed test are as follows: 1.5 m acceleration distance; 4 m at “preferred walking speed”; and 1.5 m deceleration distance. Only a 4 m walk was performed. As the participants were older adults, they were accompanied by a researcher throughout the walk to ensure their safety. For gait speed [22], the participants were categorized as slower (<1.0 m/s) or faster (≥1.0 m/s).

The Mini-Mental State Examination (MMSE) was used to measure global cognitive function [22,23]. Muscle strength was analyzed using a grip test with a digital hand dynamometer (TKK 5101 Grip-D, Takei Scientific Instruments Co. Ltd., Tokyo, Japan). During the test, the participant was positioned standing with shoulder slightly adducted with neutral rotation, elbow in 180° extension, forearm, and wrist in a neutral position and to hold the dynamometer pointing to the ground. The test was repeated twice on both the right and left hand. To obtain the best results, all participants were encouraged to perform their best during the grip test for the best result.

Body mass index (BMI) was calculated as bodyweight (kg) divided by the square of body height (m^2^).

The EuroQol-5 Dimension (EQ-D5) [24], which measures the health-related quality of life, education, smoking habits, alcohol consumption, and other risk factors, was employed as a covariate.

### 2.4. Statistical Analysis

All statistical analyses were performed using IBM SPSS V28.0 (IBM Corp., Armonk, NY, USA) and R Statistical Software (R Studio version 1.3.1093; R Foundation for Statistical Computing, Vienna, Austria). Student’s *t*-test was used to analyze the differences in baseline variables between men and women. The differences in means between the two groups were analyzed using the *t*-test for normal distribution as well as the Mann–Whitney U test for non-normal distribution. Multivariate logistic regression analysis was used to calculate the odds ratio (OR) and the 95% confidence interval (CI) of developing MCI in relation to sleep duration and gait speed.

The generalized additive model (GAM) was compared with the generalized linear model based on the Akaike information criterion. All Akaike information criterion results showed that the GAM has better model fit in our sample [25]. GAM is an extensive mathematical model of generalized linear model. This model is a nonparametric regression technique that is not restricted by linear associations [26]. The GAM replaces the linear predictor with an additive one which can model continuous data as a nonlinear smoothing function and estimate it as part of the fitting. We specified the negative binomial distribution in the GAMs to examine the dose–response relationships between sleep duration, gait speed, and the risk of developing MCI. We initially conducted univariate GAM testing and added covariates of age, sex, marital status, education, albumin levels, living status, smoking history, alcohol use, nutrition status, body mass index, and daily physical activity >3 METs for adjustment. All data were interpreted as the mean ± SD and the significance level was set at <0.05.

## 3. Results

### Participant Characteristics

There were 233 older adults (92 men, 74.1 ± 4.1 years; 141 women, 73.2 ± 4.2 years) included in the analyses. Most of the participants lived alone (9.1%), were tertiary-educated (22.3%), smoked (10.3%), drank alcohol (12.0%), had a full-time job (8.6%), and had MCI (16.8%) (Table 1).

The average participant BMI (23.2 ± 3.3 kg/m^2^) was marginally obese. Participants had healthy lifestyles as indicated by grip strength (25.0 ± 4.5 kg), MVPA engagement (14.6 ± 4.6 min/day), gait speed (1.15 ± 0.19 m/s), averaged sleep duration (361 ± 94 min/day), and a high EQ-5D index score (0.87 ± 0.02) (Table 1).

There were no statistically significant differences in the MMSE score or other variables between men and women. However, women had significantly lower education levels, alcohol drinking habits, weight, height, and grip strength than men (Table 1).

Logistic regression analyses controlling for age, sex, daily physical activity, education, current smoking status, and alcohol intake confirmed that the risk of developing MCI was related to measures of gait speed and sleep duration in the non-adjusted and multi variable-adjusted model, both with and without lifestyle adjustment (Table 2). Again, the relationships were stronger for slower gait speed and short and long sleep durations than for optimal gait speed and sleep duration. Data were adjusted for covariates and sleep time in the low gait speed group (<1 m/s), the short and long sleep duration group (<330 min and >480 min), and the group low in both variables. These showed 2.21 (1.84 after adjustment), 2.31 (1.76 after adjustment), and 5.23 (3.13 after adjustment) times greater risk of developing MCI than those in the optimal gait speed (>1 m/s) and/or sleep duration (330–480 min) groups, respectively.

Figure 1 shows the predicted relationship between sleep duration, gait speed, and the partial effect of MCI. After adjusting for age, sex, and other covariates, we found a curvilinear relationship between the two parameters and the MCI risk. The full-adjusted GAMs showed that sleep duration (20.6, *p* < 0.05) and gait speed (x^2^ = 12.7, *p* < 0.05) were significantly associated with the risk of developing MCI. A reverse J-shaped relationship between sleep duration and the risk of developing MCI was noted (degrees of freedom = 1.94) (Figure 1), with the lowest partial effects of MCI risks (0.62; standard error: 0.03) when sleeping for 455 min/night. The model also revealed a positive linear association between wake time and the risk of developing MCI (Figure 1). An exponential relationship was seen between gait speed and MCI in the full-adjusted GAM model. The gait speed showed that around 0.99 m/s has the highest partial effects on the risk of developing MCI (0.37; standard error: 0.03) (degrees of freedom = 1.75) (Figure 1).

## 4. Discussion

Our study that poor sleep and slower gait an associated with the risk of MCI. Furthermore, the combination of poor sleep and slower gait is associated with higher risk of developing MCI, particularly in older adults. This tendency is greater for individuals with poor gait and sleep than for those with only one negative variable. These results were sustained in multivariate analyses adjusted for covariates.

Our results suggest that slower gait speed (<1.0 m/s) and short (<330 min) and long (>480 min) sleep duration may be non-linearly linked to dementia risks through underlying pathways. In the older population, gait speed and/or sleep assessment may also be important for predicting cognitive impairment, including dementia.

Previous studies have suggested that abnormal sleep duration is associated with an increased risk of developing MCI [27,28] and identified a U-shaped association between sleep and cognitive function. This means both short and long sleep duration have been similarly associated with cognitive decline and an increased risk of developing MCI compared to optimal sleep duration [21,29]. In our study, however, after adjusting for all potential confounding factors, an inverted J-shaped association was found between sleep duration and the risk of developing MCI, which means that short sleep duration as compared with long sleep duration is more closely associated with increased risk of developing MCI in older adults.

Short sleep duration increases the risk of developing cardiometabolic conditions, cognitive impairment, and mortality associated with possible vascular cognitive impairment. A possible mechanism for the risk posed by sleep deprivation may be inefficient interstitial clearance of metabolic waste associated with insufficient sleep time, resulting in increased extra-neural amyloid-β accumulation. Xie et al. found that slow-wave sleep increases amyloid-β clearance, and that sleep deprivation may enhance amyloid-β accumulation. These results suggest that sleep deprivation leads to neuronal damage that could mediate cognitive decline and the onset of Alzheimer’s disease [30,31]. Moreover, sleep duration is closely related to sleep quality [32] and can disrupt circadian rhythms which control gene expression in the thalamic, hypothalamic, brainstem, locus coeruleus, and frontal lobe [33]. This may damage neurogenesis and hippocampal function, which are altered early in some neurodegenerative processes leading to cognitive impairment [34,35].

Reduced gait speed has been shown to be a strong predictor and early marker for many adverse health outcomes, such as falls, cognitive impairment, and functional decline in older adults [36,37,38]. Previous studies have shown a relationship between reduced gait speed and cognitive function, and that changes in gait speed precede cognitive decline in the course of preclinical dementia [39,40,41]. The present study found that slower gait speed was associated with the risk of developing MCI, which is consistent with previous findings. Motoric cognitive risk (MCR) is a predementia syndrome characterized by subjective complaints and objective slow gait [42]. The combination of slow gait speed and cognitive complaints has been shown to determine individuals at high risk of progressing to dementia [43,44,45]. Gait behavior is primarily controlled by frontal subcortical circuits that regulate interlinked executive function and attention and plays a key role in coordinating the functions of several neural networks of motor, sensory, and cognitive performances [46].

Gait can be affected by various factors, including pathological changes in the central nervous system. This study and previous studies have shown that gait speed is a powerful predictor of MCI risk. Gait speed can be easily measured in clinical settings, and can aid primary care clinicians more reliably to select patients at high risk of developing dementia for referral and further testing.

The relationship between sleep and physical performance is well documented. Poor sleep quality is associated with a low physical performance, including walking speed, chair rises, and standing balance, suggesting that overall physical functioning is related to sleep quality [47]. Our study adds to the evidence that the co-existence of poor sleep and slower gait is associated with higher risk of developing MCI in older adults.

These results highlight the complex interplay and synergism between sleep duration and gait abilities on the risk of developing MCI.

Few studies have examined the relationship between sleep duration and MCR, which is defined as the combination of slow gait and cognitive impairment. These found that poor sleep quality was associated with higher odds of MCR in community-dwelling older adults [48,49]. One study reported that older adults with insomnia had significantly lower gait speed and lower cognitive performance during single and dual tasks, compared to those without insomnia [50]. Stenholm et al., showed that short sleep was more often associated with mobility limitation than mid-range sleep [51]. An explanation for the relationship between sleep disturbances and insomnia and reduced mobility might be derived from excessive daytime sleepiness and fatigue [52]. Furthermore, previous studies have reported that cognitive dysfunction may lead to mobility limitation in individuals with poor sleep [53].

The shared mechanisms underlying the relationship between the co-existence of slower gait speed and poor sleep and the risk of developing MCI remain unclear. Several pathways may explain it. Various studies have reported that decreased hippocampal volume is associated with poor sleep [54], impaired cognitive function [55], and slower gait speed [56]. Inflammation due to the activation of interleukin-6 (IL-6) and C-reactive protein occurs after sleep deprivation [57] and may mediate age-related cognitive dysfunction [58]. IL-6 was also found to be related to gait ability and predictive of the risk of gait speed decline in older adults [59].

The main strength of this study is that, to the best of our knowledge, this is the first study to use the GAM model to evaluate the linear and nonlinear associations of sleep duration and gait speed and the risk of developing MCI in community-dwelling older adults.

However, the study had some limitations. It was based on cross-sectional data and could not establish a causal relationship between sleep and gait in patients with MCI. While the statistical analyses in our study are adjusted to reflect this inequality in sample size between male and female respondents, our ability to detect significant differences by gender is limited, and as the participants in this study consisted only of Koreans, the generalizability of its findings to other populations is also limited.

Another limitation is that self-reported questionnaires were used to evaluate sleep duration. Although self-reported sleep has been shown to accurately reflect objective assessments of sleep duration [60], there is a possibility that reporting may be affected by recall bias. Furthermore, time in bed may not always accurately reflect total sleep time, and this study does not distinguish between overnight sleep time and nap time throughout the day. Polysomnography can clarify whether long sleep duration was associated with sleep fragmentation, such as sleep efficiency and frequent nocturnal arousals. The strength of using self-reporting to evaluate sleep duration is that the information is easy to collect, increasing the applicability of our results to general practice.

Additionally, sleep disorder and drug therapies have altered sleep duration and circadian rhythms. Previous studies showed that obstructive sleep apnea (OSA) causes fragmented sleep [61], and altered sleep duration with reduced sleep quality might be a causal factor that results in depressive mood, mild cognitive impairment, and somatic effects [62,63,64].

Recent studies have reported the effectiveness of OSA therapies such as mandibular devices which may delay the risk of cognitive decline through OSA management [65]. However, we could not evaluate whether the participants had other sleep disorders, such as obstructive sleep apnea or restless leg syndrome, or whether they used drug therapy against sleep disorders, which could have affected the results of this study by leading to over- and underestimation of the sleep duration.

We were also unable to consider all of the potential variables such as impaired vision, falling, or fear of falling associated with slow gait speed in older adults. Previous studies have indicated that an association exists between lower gait speed and visual impairment [66], and that older adults with fear of falling seem to experience slow gait speed [67]. Hence, further studies are needed to address these challenges and provide a basis for more effective multi-modal therapies.

## 5. Conclusions

The present study suggests that the co-existence of poor sleep and slower gait is associated with a higher risk of developing MCI in community-dwelling older adults, and this relationship was found to be non-linear. In addition, gait speed was found to be positively and exponentially associated with the risk of developing MCI in the aging population, indicating that slower gait speed (<1.0 m/s) and short (< 330 min) and long (>480 min) sleep duration may be linked to dementia risks through underlying pathways.

These findings indicate that assessing gait function and sleep behaviors may be helpful in preventing MCI or cognitive decline in the aging population and that physical and sleep-related prevention of MCI may need to vary according to the age of older adults. Further research is recommended to investigate the underlying biological mechanism of the relationship by considering motor function and sleep quality and examining the synergistic association between MCI and possible moderators.

## Figures and Tables

**Figure 1 ijerph-19-07625-f001:**
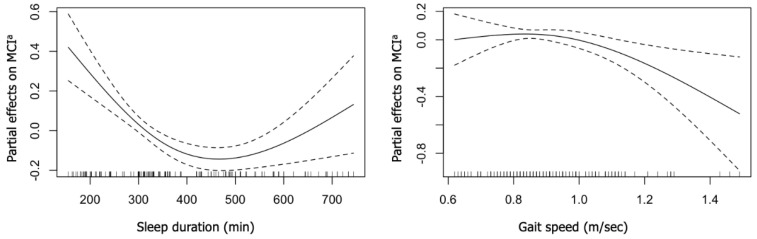
Estimated dose–response relationships of sleep duration and gait speed with risks of MCI in the total sample. Adjusted for age, sex, marital status, education, albumin levels, living status, smoking history, alcohol use, nutrition status, body mass index, and daily physical activity >3 METs. MCI^a^, mild cognitive impairment.

**Table 1 ijerph-19-07625-t001:** Selected anthropometric, social, physical and mental characteristics of participants.

Variables	Total	Men	Women
Number, n	233	92	141
Age, years	73.6 (4.3)	74.1 (4.1)	73.2 (4.2)
Weight, kg	60.1 (6.5)	67.9 (5.7) *	52.2 (6.3)
Height, m	1.61 (0.05)	1.69 (0.06) *	1.52 (0.05)
Body mass index, kg/m2	23.2 (3.3)	23.8 (3.5)	22.6 (3.0)
Tertiary education, n (%)	52 (22)	31 (33) *	21 (14)
Living alone, n (%)	21 (9.1)	8 (8.6)	13 (9.2)
Current smoker, n (%)	24 (10.3)	10 (10.8)	14 (9.9)
Current full-time job, n (%)	20 (8.6)	9 (9.7)	11 (7.8)
Alcohol drinker, n (%)	28 (12.0)	19 (20.5) *	9 (6.4)
Mild cognitive impairment, n (%)	39 (16.8)	16 (17.3)	23 (16.3)
Grip strength, kg	25.0 (4.5)	29.8 (4.9) *	20.2 (4.1)
Moderate-intensity physical activity, min/day	14.6 (4.6)	15.2 (4.2)	13.9 (5.1)
Mini-mental state examination, score	25.9 (1.9)	26.1 (2.0)	25.9 (1.9)
Gait speed, m/sec	1.15 (0.19)	1.19 (0.21)	1.12 (0.18)
Sleep duration, min	361 (94)	342 (94)	379 (98)
EQ-5D index, score	0.87 (0.02)	0.86 (0.02)	0.88 (0.01)

Values are presented as mean (SD) or n (%). * Versus men (*p* < 0.05) by chi-square and Student’s *t*-test. EQ-5D, EuroQol-5 Dimension.

**Table 2 ijerph-19-07625-t002:** Adjusted odds ratios (95% confidence intervals) for the risk of developing MCI in categories of gait speed and sleep duration in older adults.

Variables	Non Adjusted	Multi Variable Adjusted
*Gait speed*		
Optimal gait speed	Reference	Reference
Slower gait speed	2.21 (1.13–4.14)	1.84 (1.00–3.13)
*Sleep duration*		
Optimal sleep	Reference	Reference
Short and long sleep (poor sleep)	2.31 (1.03–3.54)	1.76 (1.00–3.35)
*Co-existence*		
Optimal sleep and gait speed	Reference	Reference
Slower gait or poor sleep	2.23 (1.01–3.43)	1.99 (0.83–3.47)
Slower gait and poor sleep	5.23 (2.01–7.94)	3.13 (1.93–5.14)

Odds ratios (95% confidence intervals) adjusted for age, sex, daily physical activity, education, current smoking, and alcohol consumption.

## Data Availability

Qualified researchers can obtain the data from the corresponding author (htpark@dau.ac.kr). The data are not publicly available due to privacy concerns imposed by the IRB.

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
