# Peer review of "Gait Speed and Sleep Duration Is Associated with Increased Risk of MCI in Older Community-Dwelling Adults"

_ijerph, 2022, doi:10.3390/ijerph19137625_

Round 1
Reviewer 1 Report
Thank you for the opportunity to review the manuscript. It is an interesting and significant area of research. The study conducted is scientifically sound, with sufficient rigor and has been presented in a clear manner. However, there are some areas which would require some clarification from the authors.
1. Exclusion criteria- 12 criteria have been stated for excluding participants. Perhaps it may be useful for the readers if the authors can provide some justification on why these particular exclusion criteria were selected.
2. Optimal sleep duration has been considered to to 330- 480 minutes. It would be helpful if the authors can provide an explanation or reference for choosing this cutoff.
3. The diagnosis of MCI needs to be provided in more clear detail. For example, based on which standardized criteria was MCI diagnosed etc.
4. There are minor spelling errors occasionally in the manuscript. For example line 14(abstract) has miter instead of meter/metre.
Thank you.
Author Response
Thank you for the opportunity to review the manuscript. It is an interesting and significant area of research. The study conducted is scientifically sound, with sufficient rigor and has been presented in a clear manner. However, there are some areas which would require some clarification from the authors.
Thank you very much for your variable comments. We have listed our responses to the comments point by point in the below and incorporated the comments into the revised version of the paper.
- Exclusion criteria- 12 criteria have been stated for excluding participants. Perhaps it may be useful for the readers if the authors can provide some justification on why these particular exclusion criteria were selected.
Thank you for your suggestion. We modified the information is as follows.
Line 61-70
Of the 445 participants, 233 were selected for the study and 212 were deemed ineligible based on the exclusion criteria because of other disease diagnoses or they did not meet the established criteria for inclusion (adults under the age of 65 years; heart surgery; cardiovascular disease; heart failure; congenital heart disease; pacemaker attachment; cardiovascular and cerebrovascular diseases medication; currently under treatment for musculoskeletal disorders such as osteoarthritis and rheumatoid arthritis; experience of neurological abnormalities such as chest pain; numbness; vomiting during physical activity; diagnosed with a specific disease from a specialist and have been recommended to perform physical activity only under the guidance of a specialist or prohibited to perform physical activity; feeling pain in the chest when resting or arrhythmia; missing data; did not agree in writing to participate in the study)
- Optimal sleep duration has been considered to to 330- 480 minutes. It would be helpful if the authors can provide an explanation or reference for choosing this cutoff.
We agree with your suggestion. We have added explanation of how to determine the cut-off points for optimal sleep duration.
Line 84-90
2.2. Sleep duration
The cut-off value of 330-480 min for optimal sleep duration was defined as the point at which MCI risk begins to increase from zero based on our results of the GAM analysis. Thus, the participants were divided into three groups according to their sleep duration (short, <330 min; optimal, 330-480 min; long, >480 min). This result consistent with previous study that reported with individuals having <6 or ≤5 hours or > 8 or 9 hours of sleep showing an increased risk of dementia or MCI [20, 21].
- The diagnosis of MCI needs to be provided in more clear detail. For example, based on which standardized criteria was MCI diagnosed etc.
Thank you for appropriate comments. We revised the detailed information as follows
Line 71-77
Based on the methodology outlined in a previously published paper MCI diagnosis [17], diagnostic assignment was made by the study’s neurologist based on data from the clinical examination, rating scales and neuropsychological testing. With regard to the later, subjects were classified as meeting criteria for amnestic MCI if they per-formed at least 1.5 standard deviations below the age/intelligence quotient adjusted mean on at least 1 measure of memory functioning, with preserved general cognitive functioning [17].
- There are minor spelling errors occasionally in the manuscript. For example, line 14(abstract) has miter instead of meter/metre.
Thank you for your suggestion. We have corrected the typo.
Line 14
meter
Reviewer 2 Report
Dear Author,
1 list the inclusion criteria;
2 result: attention, according to the BMI scale a score between 25 and 30 identifies an overweight patient and not an obese;
3 specify in the discussion that the sample is made up of Koreans: there are too many inter-individual variables between the different populations that could limit the results of your study;
4 sleep duration: it was not objectively measured with a test (like Home Sleeping Test) but each patient answered a question (the sample consists of elderly patients, the answer is not reliable); explain why you don’t use a specific test;
5 sleep disorder drugs may have altered sleep hours and circadian rhythms: explain why you have not evaluated drug therapy against sleep disorders in the sample;
6 the sample has a higher number of women (141 vs 92 men): there may be some differences related to sex to consider;
7 the slowed gait could be linked to other unanalyzed variables such as vision, which often decreases in the elderly;
8 respiratory function and the risk of obstructive sleep apnea which could be the cause of the reduced number of hours of sleep are not considered;
9 The discussion section needs an improvement , following my previous comments. I would suggest the following references, but there are also more to use:
· -A new design of mandibular advancement device (IMYS) in the treatment of obstructive sleep apnea (10.1080/08869634.2022.2041271);
· -Sleep quality: An evolutionary concept analysis ( doi: 10.1111/nuf.12659 );
· -Poor sleep quality and physical performance in older adults (doi: 10.1016/j.sleh.2020.10.002);
· -Associations between gait speed and well-known fall risk factors among community-dwelling older adults (doi: 10.1002/pri.1743)
Author Response
1 list the inclusion criteria;
Thank you for your suggestion. Our participants were recruited from the four community health center and institution, Busan city, Korea and community-dwelling older adults aged 65 or more. We included a total 445 individuals who agreed to participate and completed our screening survey. We have modified the following description under Participants in the Materials and Methods section.
Line 57-60
2.1. Participants
our participants were recruited from the four community health center and institution, Busan city, Korea and community-dwelling older adults aged 65 or more. We included a total 445 individuals who agreed to participate and completed our screening survey.
2 results: attention, according to the BMI scale a score between 25 and 30 identifies an overweight patient and not an obese;
Thank you for your comment. WHO reported the increased health risks associated with obesity occur in people with lower BMIs in the Asia-Pacific region. In 2000, WHO recognized the need for different BMI standards for the Asia-Pacific region and suggested new standards for the Asia-Pacific region (https://apps.who.int/iris/handle/10665/206936). The subjects of this study were all Koreans. Therefore, based on the criteria established by the Korea Centers for Disease Control and Prevention (https://health.kdca.go.kr/healthinfo/biz/health/gnrlzHealthInfo/gnrlzHealthInfo/gnrlzHealthInfoView.do?cntnts_sn=5292) and the Korean Society for the Study of Obesity(http://general.kosso.or.kr/html/?pmode=obesityDiagnosis), cases with BMI of 25 or higher were defined as obesity in this study.
3 specify in the discussion that the sample is made up of Koreans: there are too many inter-individual variables between the different populations that could limit the results of your study;
Thank you for your suggestion. We have added it as a study limitation
Line 254-257
- Discussion
Line 254-257
While the statistical analyses in our study are adjusted to reflect this inequality in sample size between male and female respondents, our ability to detect significant differences by gender is limited, and as the participants of this study consisted only of Koreans, the generalizability of these findings to the other population are also limited.
4 sleep duration: it was not objectively measured with a test (like Home Sleeping Test) but each patient answered a question (the sample consists of elderly patients, the answer is not reliable); explain why you don’t use a specific test;
Thank you for your helpful suggestion. According to your suggestion, self-reported sleep duration has been shown to reflect objective assessments of sleep accurately, reporting may be influenced by recall bias. We have added the self-report sleep duration about recall bias as limitations in the Discussion section.
Line 258-265
- Discussion
Although self-reported sleep has been shown to accurately reflect objective assessments of sleep duration [60], it is a possibility that reporting may be affected by recall bias. Furthermore, time in bed may not always accurately reflect total sleep time, and this study does not distinguish between overnight sleep time and nap time throughout the day. Polysomnography would clarify whether long sleep duration was associated with sleep fragmentation, such as sleep efficiency and frequent nocturnal arousals. The strength of using self-report to evaluation sleep duration is that the information is easy to collect, increasing the applicability of our results to general practice.
5 sleep disorder drugs may have altered sleep hours and circadian rhythms: explain why you have not evaluated drug therapy against sleep disorders in the sample;
Thank you for your appropriate recommendation. According to your suggestion, sleep disorder drugs may have altered sleep hours and circadian rhythms. However, we could not evaluate sleep-related drugs and sleep disorder in our survey. We have added these limitations in the Discussion section.
Line 265-272
- Discussion
Additionally, sleep disorder and drug therapy have altered sleep duration and circadian rhythms. A previous studies showed that obstructive sleep apnea causes fragmented sleep [61], and altered sleep duration with reduced sleep quality might be a causal factor that results in depressive mood, emotional, cognitive, and somatic effects [62, 63]. However, we could not evaluate whether the participants had other sleep disorders, such as obstructive sleep apnea, restless legs syndrome, and drug therapy against sleep disorders, which could have affected the results of this study by leading to over- and underestimation of the sleep duration.
6 the sample has a higher number of women (141 vs 92 men): there may be some differences related to sex to consider;
When we run the logistic regression analyses, we put 'sex' in the model as a confounder in the adjusted model. We had specified it in the table footnote but left out it in the text by accident. We have corrected the sentence.
Nevertheless, as your suggestion, we also added the limitation in the discussion.
Line 147-150
- Result
Logistic regression analyses controlling for age, sex, daily physical activity, education, current smoking status, and alcohol intake confirmed that both with and without lifestyle adjustment, the risk of MCI was related to measures of gait speed and sleep duration in the non-adjusted and multi variable-adjusted model (Table 2).
Line 254-257
While the statistical analyses in our study are adjusted to reflect this inequality in sam-ple size between male and female respondents, our ability to detect significant differ-ences by gender is limited, and as the participants of this study consisted only of Kore-ans, the generalizability of these findings to the other population are also limited
7 the slowed gait could be linked to other unanalyzed variables such as vision, which often decreases in the elderly;
Thank you for your suggestion. As you suggest we also think there are many factors related to the slow gait speed in older adults. We have added these limitations in the Discussion section.
Line 272-276
- Discussion
We were also unable to consider all of the potential variables such as vision, fall, and fear of falling associated with slow gait speed in older adults. Previous studies have indicated association between lower gait speed and visual impairment [64], and older adults with fear of falling seems to slow gait speed [65]. We plan to examine this issue in future studies.
8 respiratory function and the risk of obstructive sleep apnea which could be the cause of the reduced number of hours of sleep are not considered;
Thank you for your appropriate recommendation. According to your suggestion, sleep disorder drugs may have altered sleep hours and circadian rhythms. However, we could not evaluate sleep-related drugs and sleep disorder in our survey. We have added these limitations in the Discussion section.
Line 265-272
- Discussion
Additionally, sleep disorder and drug therapy have altered sleep duration and circadian rhythms. A previous studies showed that obstructive sleep apnea causes fragmented sleep [61], and altered sleep duration with reduced sleep quality might be a causal factor that results in depressive mood, emotional, cognitive, and somatic effects [62, 63]. However, we could not evaluate whether the participants had other sleep disorders, such as obstructive sleep apnea, restless legs syndrome, and drug therapy against sleep disorders, which could have affected the results of this study by leading to over- and underestimation of the sleep duration.
9 The discussion section needs an improvement, following my previous comments. I would suggest the following references, but there are also more to use:
- -A new design of mandibular advancement device (IMYS) in the treatment of obstructive sleep apnea (10.1080/08869634.2022.2041271);
- -Sleep quality: An evolutionary concept analysis ( doi: 10.1111/nuf.12659 );
- -Poor sleep quality and physical performance in older adults (doi: 10.1016/j.sleh.2020.10.002);
- -Associations between gait speed and well-known fall risk factors among community-dwelling older adults (doi: 10.1002/pri.1743)
Thank you for your helpful recommendation. We have enhanced the discussion section according to your comments by referring to the references.
Line 209-211
- Discussion
Reduced gait speed has been shown to be a strong predictor and early marker for many adverse health outcomes, such as falls, cognitive impairment, and functional decline in older adults [36-38].
Line 226-229
- Discussion
The relationship between sleep and physical performance is well documented. Poor sleep quality was significantly more likely to have a low physical performance including walking speed, chair rises, and standing balance, suggests that overall physical functioning is related to sleep quality [47].
Round 2
Reviewer 2 Report
Dear Authors,
the paper was corrected in all sections.
I suggested to include this papers but i cannot find in the paper:
-A new design of mandibular advancement device (IMYS) in the treatment of obstructive sleep apnea Cranio. 2022 Feb 16;1-8. doi: 10.1080/08869634.2022.2041271.
-The prevalence of obstructive sleep apnea in mild cognitive impairment: a systematic review BMC Neurol. 2019 Aug 15;19(1):195. doi: 10.1186/s12883-019-1422-3.
Best regards
Author Response
The Paper was corrected in all sections. I suggested to include this papers but i cannot find in the paper:
-A new design of mandibular advancement device (IMYS) in the treatment of obstructive sleep apnea (10.1080/08869634.2022.2041271);
-The prevalence of obstructive sleep apenea in mild cognitive impairment: a systematic review BMC Neurol. 2019 Aug 15:19(1):195. doi: 10.1186/s12883-019-1422-3.
Thank you very much for your comments. As your suggestion, we added the related contents and cited two references.
Line 266-272
Additionally, sleep disorder and drug therapy have altered sleep duration and circadian rhythms. A previous studies showed that obstructive sleep apnea (OSA) causes fragmented sleep [61], and altered sleep duration with reduced sleep quality might be a causal factor that results in depressive mood, mild cognitive impairment, and somatic effects [62, 63, 64]. Recent studies have reported the effectiveness of the OSA therapies such as mandibular devices which may delay the risk of cognitive decline through OSA management [65].
Added reference
- Mubashir, T.; Abrahamyan, L.; Niazi, A.; Piyasena, D.; Arif, A, A.; Wong, J., The prevalence of obstructive sleep apnea in mild cognitive impairment: a systematic review. BMC Neurol 2019, 19, (1), 195.
- Ciavarella, D.; Campobasso, A.; Suriano, C.; Lo Muzio, E.; Guida, L.; Salcuni, F.; Laurenziello, M; Illuzzi, G; Tepedino, M., A new design of mandibular advancement device (IMYS) in the treatment of obstructive sleep apnea. Cranio 2022, 1-8.